# Current Data and Future Perspectives on Patients with Atrial Fibrillation and Cancer

**DOI:** 10.3390/cancers15225357

**Published:** 2023-11-10

**Authors:** Leonardo De Luca, Massimiliano Camilli, Maria Laura Canale, Raffaella Mistrulli, Federico Andreoli, Guido Giacalone, Fabio Maria Turazza, Domenico Gabrielli, Irma Bisceglia

**Affiliations:** 1Department of Cardio-Thoracic and Vascular Medicine and Surgery, Division of Cardiology, A.O. San Camillo-Forlanini, 00152 Rome, Italy; dgabrielli@scamilloforlanini.rm.it (D.G.); ibisceglia@scamilloforlanini.it (I.B.); 2Department of Cardiovascular and Pulmonary Sciences, Catholic University of the Sacred Heart, 00168 Rome, Italy; massimiliano.camilli@guest.policlinicogemelli.it; 3Division of Cardiology, Azienda USL Toscana Nord-Ovest, Versilia Hospital, 55041 Lido di Camaiore, Italy; marialaura.canale@uslnordovest.toscana.it; 4Clinical and Molecular Medicine Department, Sapienza University of Rome, 00185 Roma, Italy; raffaella.mistrulli@uniroma1.it (R.M.); federico.a2@outlook.it (F.A.); guido.giacalone@uniroma1.it (G.G.); 5Cardiology Unit, Fondazione IRCCS Istituto Nazionale dei Tumori, 20133 Milan, Italy; fabio.turazza@istitutotumori.mi.it

**Keywords:** atrial fibrillation, cancer, anticoagulation

## Abstract

**Simple Summary:**

Atrial fibrillation is a very common comorbidity in cancer patients. Both cancer and cancer treatment increase the risk of developing new AF, resulting in increased morbidity and mortality. Heart rate and rhythm control, together with anticoagulant therapy, are the mainstays of atrial fibrillation treatment, even in patients with cancer. However, treatment adjustments may be necessary due to drug interactions with concomitant chemotherapy. In addition, cancer and advanced age increase the risk of both thromboembolism and hemorrhage. The risk of these complications is further increased by concomitant cancer therapy, frailty, poor nutrition status, and coexisting geriatric syndromes. In this review, we address the complex mechanisms linking arrhythmia to cancer and the difficult therapeutic challenges faced by oncologists and cardiologists in best managing these two conditions.

**Abstract:**

Atrial fibrillation (AF) is an increasingly recognized comorbidity in patients with cancer. Indeed, cancer patients have a significantly higher incidence of AF than that observed in the general population. A reciprocal relationship between these two diseases has been observed, as much as some assume AF to be a marker for occult cancer screening, especially in older adults. The pathophysiological mechanisms are many and varied, including the underlying pro-inflammatory state, specific treatments (chemo- and radiotherapy), and surgery. The therapeutic management of patients with cancer and AF involves the same rhythm and frequency control strategies as the general population; however, the numerous interactions with chemotherapeutics, which lead to a significant increase in side effects, as well as the extreme fragility of the patient, should be considered. Anticoagulant therapy is also a complex challenge to address, as bleeding and stroke risk scores have not been fully assessed in this subpopulation. Furthermore, in large studies establishing the efficacy of direct oral anticoagulants (DOACs), cancer patients have been underrepresented. In this review, we elaborate on the mechanisms linking AF to cancer patients with a particular focus on the therapeutic challenges in this population.

## 1. Introduction and Epidemiology: Atrial Fibrillation in Cancer Patients

The improvement in cancer patients’ prognoses and therefore the aging of this population, as well as the introduction of targeted therapies, have exponentially increased the incidence of cardiac arrhythmias seen in oncology and hematology wards [1,2,3]. In particular, AF, a leading cause of thrombotic morbidity and overall cardiovascular (CV) mortality, is the most common sustained arrhythmia in the general population and has been revealed to be more common in patients with malignancies [1,2,3,4], reaching an incidence of 30% in the available studies [5,6,7,8]. In this setting, the prevalence seems extremely variable in the literature, depending on the age of the population examined, pre-existing risk factors, the type of primitive cancer, previous oncologic surgery, and the chemotherapy schemes instituted [9,10,11,12,13]. Indeed, the risk of AF is higher in subjects older than 65 years with known CV disease [14], as well as in those patients affected by all hematologic malignancies, including lymphoma, leukemia, and multiple myeloma, rather than solid tumors [15]. Moreover, higher cancer stages and grades at diagnosis raise the risk of AF, even suggesting a systemic effect of advanced cancer itself on the heart [7]. Of importance, post-operative AF is the most frequent form of sustained arrhythmia in cancer patients. Its prevalence ranges from 16 to 46% for cardiothoracic surgery and 0.4–12% in non-cardiothoracic surgery, increasing the post-operative mortality, the hospitalization length, and intensive care unit admissions [16,17]. AF may therefore represent an additional determinant of malignancies’ prognoses and a challenge for the therapeutic management of cancer patients [18,19]. The aim of this review is thus to elucidate novel etiological aspects subtending the AF occurrence in this population, give advice on management aspects, and shed light on future research scopes in this expanding field of cardio-oncology.

## 2. Risk Factors and Pathogenesis of Atrial Fibrillation in Cancer Patients

To date, inflammation-related oxidative stress in cancer is believed to cause electrical and anatomical changes that predispose and maintain AF, including through fibrosis. C-reactive protein (CRP), interleukins (ILs), in particular IL-2, IL-6, and IL-8, macrophage migration inhibition factor (MIF), and tumor necrosis factor alpha are all elevated in AF and cancer patients [20]. Increased inflammatory markers can lead to autonomic dysfunction, electrolyte imbalances, structural alterations of the heart, and electrical remodeling. Alterations in calcium hemostasis and connexins can cause a number of atrial conduction abnormalities, including AF [21]. A causal role of inflammation in AF has been suggested by studies showing the increased activation of the NLRP3 inflammasome (NACHT, LRR, and PYD domain containing protein 3) in AF [22]. The NLRP3 inflammasome mediates caspase-1 activation and interleukin-1β release in immune cells, and this interleukin is increased in cancer patients, also promoting AF onset. The neoplasm-related pro-inflammatory state also includes an increase in reactive oxygen species (ROS), which are a by-product of increased cell metabolism and can promote atrial fibrosis and the remodeling of the extracellular matrix of the atrium through the activation of metalloproteinases [23,24]. In summary, inflammation plays a central role in the development and progression of cancer and thus, subsequently, in the trigger or maintenance of AF.

More research is needed to determine the role of anti-inflammatory therapies in cancer prevention, as well as in the prevention of AF [25]. Many anticancer drugs have been associated with an increased risk of AF both in terms of incident and recurrent AF. Cancer drug-induced AF may occur shortly after treatment (cisplatin or gemcitabine) or weeks or months after starting treatment, as in the case of ibrutinib [25]. Tyrosine kinase inhibitors (TKIs), immunomodulators, like interleukin-2 (IL-2), antimetabolites, like 5-fluorouracil and gemcitabine, HER-2/Neu receptor blockers, alkylating agents, anthracyclines, and antimicrotubular agents have all been related to the development of new-onset AF. Ibrutinib is a Bruton TKI used to treat a variety of B-cell malignancies. It is the TKI most linked to an increased risk of AF, with up to 16% of patients developing AF after starting therapy (Table 1) [26]. The off-target inhibition of other tyrosine kinases in cardiac myocardial cells may be the mechanism underlying the development of AF in patients [27]. Ibrutinib, for example, has been shown to inhibit C-terminal Src kinase. A knockout mouse model lacking C-terminal Src kinase was found to induce left atrial enlargement, fibrosis, and inflammation, resulting in increased AF. Furthermore, ibrutinib may cause AF by producing ROS [28]. Immune checkpoint inhibitors are also commonly used to treat specific types of cancer and have been linked to cardiotoxicity, myocarditis, and AF caused by altered inflammation. Surgical procedures, such as lung resections or other extensive operations, are also often followed by perioperative AF. In a cohort of 13,906 patients undergoing lung resections for lung cancer, perioperative AF occurred in 12.6% of patients [29]. Perioperative AF appears to be more frequent in patients with advanced ages and stages of cancer who have cardiovascular comorbidities and who undergo extensive resections [30]. Furthermore, high adrenergic states following cancer surgery may induce or worsen AF [31]. Infection, anemia, hypoxia, pleurisy, pericarditis, and cardiomyopathy are all potential complications of cancer and cancer treatment and are all are potential triggers of AF [32]. More rarely, AF may be triggered by the metastatic involvement of the heart [25]. The most common neoplasms associated with cardiac metastases are lung cancer, lymphoma, breast cancer, leukemia, stomach cancer, and melanoma [33]. Cardiac metastases mostly appear in elderly patients who already have disseminated cancer disease. Tumors may reach the heart via the lymphatic or intravenous route, or by direct extension, and the sites most affected are the pericardium or epicardium [34]. There is a growing understanding of the shared risk factors that may be responsible for the development or progression of cancer and AF.

Modifiable risk factors, such as hypertension and obesity, continue to be underdiagnosed and undertreated in cancer patients [35]. To improve the long-term outcomes in cancer patients, early diagnosis via standardized risk-based screening and the management of these conditions in accordance with general ESC guidelines are recommended [10] (Figure 1).

## 3. Management of Atrial Fibrillation in the Setting of Cancer

### 3.1. Rate and Rhythm Control

Although the management of AF in patients with cancer should follow the 2020 European Society of Cardiology (ESC) guidelines on AF and the “ABC pathway” approach should be applied, there are some exceptions in which treatment modifications should be considered [1,10].

Among the rate-control drugs, beta-blockers are preferred, especially if the cancer therapies have a potential cardiac dysfunction risk. Calcium channel blockers (diltiazem and verapamil) should be avoided if possible due to drug–drug interactions and negative inotropic effects. The same applies to digoxin, which is to be considered a second choice [26].

The decision to convert AF to the sinus rhythm (rhythm control) is made individually for each patient. For older adults, who are especially vulnerable to the side effects of antiarrhythmic medications, there is less emphasis on rhythm control. Rhythm control may be indicated in patients who are significantly symptomatic from AF or whose AF is difficult to rate-control [36]. To convert AF to the sinus rhythm, both electrical and pharmacologic methods can be used. For unstable patients (altered mental status, hypotension, chest pain, or hypoxia attributed to arrythmia), emergency electrical cardioversion is the first-line therapy. Flecainide and propafenone are antiarrhythmic medications that are frequently used for pharmacologic cardioversion. However, many older adults, including those with cancer, have underlying structural heart disease, which restricts the use of these therapies in this group due to its increased pro-arrhythmic effects [37].

Although amiodarone is effective at maintaining the sinus rhythm, it has greater toxicities than other antiarrhythmics used in AF. There is a strong temporal relationship between therapy with taxanes, such as paclitaxel and docetaxel, used for the treatment of many cancers, such as breast and lung cancer, and the development of severe skin and mucosal toxicity due to the reduced clearance of taxanes in patients taking amiodarone [38]. Amiodarone has also been shown to increase the adverse effects of radiation on the skin and mucous membranes [39]. In older adults with normal QTc intervals, sotalol, a class III antiarrhythmic agent, may be a good choice for maintaining the sinus rhythm [31]. However, several anticancer treatments may contribute to QTc prolongation, which can lead to life-threatening ventricular arrhythmias [40]. Kinase inhibitors, such as dasatinib and ruxolitinib, used to treat chronic myeloid leukemia and myelofibrosis, may cause QTc interval prolongation. Arsenic trioxide, which is used to treat promyelocytic leukemia, may also cause QTc interval prolongation. Some anti-emetic drugs, such as ondansetron, which is commonly used in cancer patients to prevent and treat nausea, may also contribute to QTc prolongation [29].

AF in patients treated with ibrutinib can be particularly challenging to manage given the multiple drug–drug interactions between ibrutinib and many agents used for rate or rhythm control in AF (Table 1).

### 3.2. Non-Pharmacological Management of AF in the Setting of Cancer

The possibility of the ablation of atrial fibrillation should be discussed in selected patients with heart failure (HF) and uncontrolled symptoms, taking into account their cancer status and prognosis [41].

In a retrospective study, the ablation of AF in patients with cancer in the preceding 5 years or with exposure to anthracyclines and/or thoracic radiation at any time prior to index ablation was analyzed in comparison with patients with no history of cancer. The primary outcome was freedom from atrial fibrillation (with or without antiarrhythmic drugs, or the need to repeat catheter ablation at 12 months after the first procedure of ablation). Freedom from atrial fibrillation at 12 months was not different in the two comparison groups, and the need to repeat the ablation was also similar between the groups (20.7% vs. 27.5%, *p* < 0.29) [42]. In addition, there were no differences in the safety endpoints between the groups with regard to the risk of bleeding. However, data on ablation in cancer patients are still limited. Finally, if the above-mentioned strategies fail to control the AF, then AV node ablation with permanent pacing should be considered to alleviate the symptoms and hemodynamic effects of refractory AF [43].

### 3.3. Anticoagulant Treatment

(a)Risk–benefit decision regarding anticoagulation: ischemic and bleeding risk

Anticoagulant therapy is a complex challenge, as cancer patients present both high thrombotic risk and high hemorrhagic risk. According to the ESC guidelines, the therapeutic decision should be based on both the CHA2DS2-VASc (congestive heart failure, hypertension, age ≥ 75 years (2 points), diabetes mellitus, stroke (2 points), vascular disease, age 65–74 years, sex category (female)) score and on hemorrhagic risk scores, such as the HAS-BLED (hypertension, abnormal renal and liver function, stroke, bleeding, labile international normalized ratio, elderly, drugs or alcohol), although these have not been validated in cancer patients (Table 2) [41,42,43,44]. In a retrospective cohort study including 2,435,541 adults hospitalized with AF, the predictive value of the CHA2DS2-VASc score was lower in patients with cancer than in those without. In another retrospective cohort study, patients with AF and cancer and with AF without cancer were compared. Both groups had CHA2DS2-VASc scores of from 0 to 2 and were not receiving anticoagulation upon the diagnosis of cancer or at the date of inclusion in the study. The primary outcome was the risk of arterial thromboembolism (ischemic stroke, transient ischemic attack, or systemic arterial thromboembolism) at 12 months. The 12-month cumulative incidences of arterial thromboembolism were 2.13% (95% CI: 1.47–2.99) in 1411 AF patients with cancer and 0.8% (95% CI: 0.56–1.10) in 4233 AF patients without cancer (HR: 2.70; 95% CI: 1.65–4.41). The risk was higher in men with CHA2DS2-VASc scores = 1 and in women with CHA2DS2-VASc scores = 2 (HR: 6.07; 95% CI: 2.45–15.01) [45]. Although cancer is not mentioned in the CHA2DS2-VASc score, the latter is associated with a propensity for thrombosis [46]. Regarding the assessment of the bleeding risk, the HAS-BLED was quite accurate [47], although the HEMORR2HAGES score also includes a history of malignancy and thrombocytopenia in the risk assessment (Table 3) [48]. The latter is an important finding, as it has been shown that platelets < 100,000 × 10^9^/L increase the risk of bleeding for cancer patients taking anticoagulants and tumors together with cancer treatments may cause thrombocytopenia [33,49]. Farmakis et al. proposed an alternative approach for risk stratification that includes the following acronyms: T (thrombotic risk), B (bleeding risk), I (drug interactions), and P (patient access and preferences) [50]. This algorithm guides the clinician in adopting an appropriate therapy based on a comprehensive assessment of all aspects of the cancer patient (Figure 2).

The TBIP approach has been adopted by 2022 cardio-oncology guidelines [1], and it thus provides a more comprehensive and cancer-focused approach to anticoagulation in cancer patients with AF. However, prospectively validated clinical scores to guide anticoagulation in cancer patients with AF are badly needed.

(b) Choice of anticoagulant therapy.

Once the need for anticoagulant therapy is established, it is necessary to evaluate which drug is most appropriate for the specific patient. Vitamin K antagonists (VKAs) are recognized to be effective at reducing the thromboembolic risk in patients with atrial fibrillation and cancer; despite this, compared to cancer-free controls, cancer patients who take warfarin—whether for NVAF or venous thromboembolism (VTE)—have worse anticoagulation management and worse outcomes, including a six-fold increase in bleeding rates.

Additionally, a large reduction in the time in the therapeutic range is linked to the development of cancer in those using long-term warfarin, especially within the first six months following the cancer diagnosis [51]. Furthermore, its use in these patients is complicated by drug–drug interactions with chemotherapy drugs that occur through several mechanisms, including the induction or inhibition of cytochrome P450 isozymes, the displacement of binding from plasma proteins, and alterations in the vitamin K status. Despite these difficulties, warfarin has long been the drug of choice for NVAF anticoagulation.

Low-molecular-weight heparins (LMWHs) have not been proven to be effective in preventing stroke or systemic embolism in AF and cancer, and their use is only justified by their demonstrated efficacy and safety in venous thromboembolism (VTE) [1]. Their use is often limited to the perioperative-bridging period for patients on warfarin.

It is debatable whether data supporting the use of LMWHs as perioperative-bridging agents can be extended to their long-term use, as chemotherapy regimens last many months, and thus it is difficult to predict whether the long-term use of LMWHs is safe and effective.

No specifically designed randomized controlled trial has looked at the use of non-vitamin K antagonist oral anticoagulants (NOACs) for AF in cancer patients. Large observational studies and post hoc analyses of pivotal trials utilizing NOACs in AF patients indicate that they are safe and at least as effective as VKAs in patients with AF and active cancer.

A minority of patients with a history of cancer (640 out of 14,264) were enrolled in the ROCKET AF trial, with the most common types of malignancies being prostate, colorectal, and breast cancer. There were no significant differences between rivaroxaban and warfarin in terms of the relative efficacy and safety between patients with and without a history of cancer. The risk of ischemic events was not affected by a history of malignancy, although it did raise the risk of bleeding and non-cardiovascular death [52,53].

A history of cancer was present in 6.8% of participants in the ARISTOTLE trial. A history of cancer was not substantially related with major bleeding, mortality, stroke, or systemic embolism. Apixaban was as effective as warfarin at preventing stroke and systemic embolism in patients with and without a history of cancer, and its safety profile was comparable to that of warfarin [54].

A minority (5.5%) of patients in the ENGAGE AF-TIMI 48 study had a new or recurrent cancer diagnosed, with the gastrointestinal tract, prostate, and lung being the most common sites. Malignancy per se was associated with a higher risk of overall mortality and severe bleeding, but not with stroke or systemic embolism. In AF patients who develop cancer, edoxaban maintains its efficacy and safety profile, making it a potentially more useful treatment choice [55].

NOACs showed a better safety profile than warfarin in patients with underlying malignancies and AF, according to a large retrospective American database investigation. Warfarin was associated with greater death rates in addition to a higher risk of hemorrhagic stroke [56].

To confirm the safety and effectiveness of NOACs in patients with active malignancy and AF, an administrative dataset was examined. NOAC users had decreased or equivalent rates of bleeding, stroke, and incident VTE compared to warfarin users [53].

An additional study of 40,271 individuals with AF and cancer using retrospective data from Medicare and other commercial claim databases revealed that apixaban was associated with a lower risk of stroke/systemic embolism and significant bleeding compared to warfarin, although dabigatran and rivaroxaban exhibited equivalent hazards [57]. According to a recent meta-analysis, NOACs were linked to a significantly lower rate of serious bleeding complications and thromboembolic events in patients with cancer and AF compared to VKAs [58].

NOACs, with apixaban being the best of those examined, demonstrated a decreased incidence of stroke/systemic embolism, VTE, all-cause death, and significant bleeding in AF patients with cancer compared with warfarin, according to a network meta-analysis [59].

At the MD Anderson Cancer Center, 1133 patients with current malignancies and AF were included in a recent single-institution retrospective analysis. The results in terms of the cerebrovascular accident, gastrointestinal bleeding, and cerebral hemorrhage of NOACs versus VKAs were compared using propensity score matching. The study revealed that patients with active malignancies had equivalent risks for cerebrovascular accident, gastrointestinal bleeding, and cerebral hemorrhage when given NOACs instead of warfarin for AF [60].

According to a Surveillance, Epidemiology, and End Results cancer registry database analysis, similar risks of stroke, systemic embolism, and severe bleeding have been observed in older persons with cancer and AF who were exposed to NOACs or warfarin. In comparison to warfarin, NOAC use was linked to a decreased risk of death from all causes and a similar risk of cardiovascular death [61].

Although the use of NOACs for AF in cancer patients grew from 2010 to 2016, there is still a significant percentage of patients with AF and cancer who are not taking anticoagulants [62].

According to recent ESC guidelines on cardio-oncology, the use of NOACs in cancer patients with AF is broadly accepted in light of previous findings, even if a clear prospective evaluation is lacking. NOACs should be considered for stroke prevention instead of LMWHs and VKAs in patients without significant drug–drug interactions, mechanical heart valves, or moderate-to-severe mitral stenoses [51].

Similarly, the International Society on Thrombosis and Haemostasis has recommended that specific decisions regarding anticoagulation for patients with cancer and AF should be made after considering the relative risks of stroke and bleeding. If there are no substantial interactions with oncological medications in patients who started anticoagulation prior to receiving anticancer treatment, then the therapy should not be changed. If there are no substantial drug–drug interactions, then NOACs should be chosen over VKAs or low-molecular-weight heparins in patients with newly diagnosed AF receiving chemotherapy. Patients with gastrointestinal neoplasms or other gastrointestinal tract conditions that increase the bleeding risk are the exception [51].

Although several reports point to the efficacy and safety of NOACs in cancer patients with AF, RCTs are needed to confirm these results [63].

Individuals with active cancer constitute a challenging patient population that requires extra attention. Oral anticoagulant therapy in cancer patients may be hampered by other factors, like drug–drug interactions, renal impairment, and thrombocytopenia [64]. Drug interactions are not limited to anticancer agents but may include other supportive care drugs (i.e., antiemetics, opioids, etc.) that must also be taken into consideration [65].

The guidelines recommend the percutaneous closure of the left appendix (LAA) in patients with life expectancies of more than one year who are at high thromboembolic and hemorrhagic risk and in whom anticoagulation is contraindicated. However, LAA devices are only used in a very select few cancer patients due to implant-related complications—including device-related thrombosis—and the lack of prospective data on this specific population [66].

In a retrospective analysis of patients referred for LAA closure, individuals with active cancer had a higher probability of having an in-hospital transient ischemic attack or stroke than patients with no active cancer or prior history of cancer. The incidence of the 30-day and 180-day readmission outcomes, as well as the composite in-hospital outcome rate (in-hospital death, ischemic stroke/transient ischemic attack, systemic embolism, bleeding requiring blood transfusion, pericardial effusion/cardiac tamponade treated with pericardiocentesis or surgically, and the removal of an embolized device), did not differ significantly between the groups [67].

Active cancer patients are likely to benefit from a closer follow-up plan with regular re-evaluations given the rapidly changing clinical scenario. A multidisciplinary management approach that considers individual bleeding and thrombotic risks, drug–drug interactions, patient preferences, and routine clinical evaluation is necessary to identify the appropriate anticoagulation strategy for cancer patients [67]. As previously stated, the algorithm proposed by Pastori et al. (Figure 2) could represent a useful guide for the management of this complex group of patients. The safety and efficacy of NOACs for stroke prevention in cancer patients with AF are supported by accumulating research, making them a viable anticoagulation therapy option (Table 4).

## 4. Conclusions

AF is a very common comorbidity in cancer patients, as there are several mechanisms that can trigger it or make it worse. Rate control is frequently preferred over rhythm control in cancer patients due to the higher prevalence of side effects associated with anti-arrhythmic drugs and their numerous drug–drug interactions with various chemotherapeutic agents.

Anticoagulation risk–benefit ratio decisions and anticoagulant drug selection remain difficult challenges. This population is predisposed to thromboembolic and hemorrhagic complications. The current risk scores used in the general population have not been validated in this subgroup and do not always provide a true estimate of the risk. Although there is substantial evidence in favor of DOACs, they are currently underutilized in favor of LMWHs and VKAs, which should be considered a second choice. Close follow-up remains a key issue, given the rapidly changing clinical scenario.

## Figures and Tables

**Figure 1 cancers-15-05357-f001:**
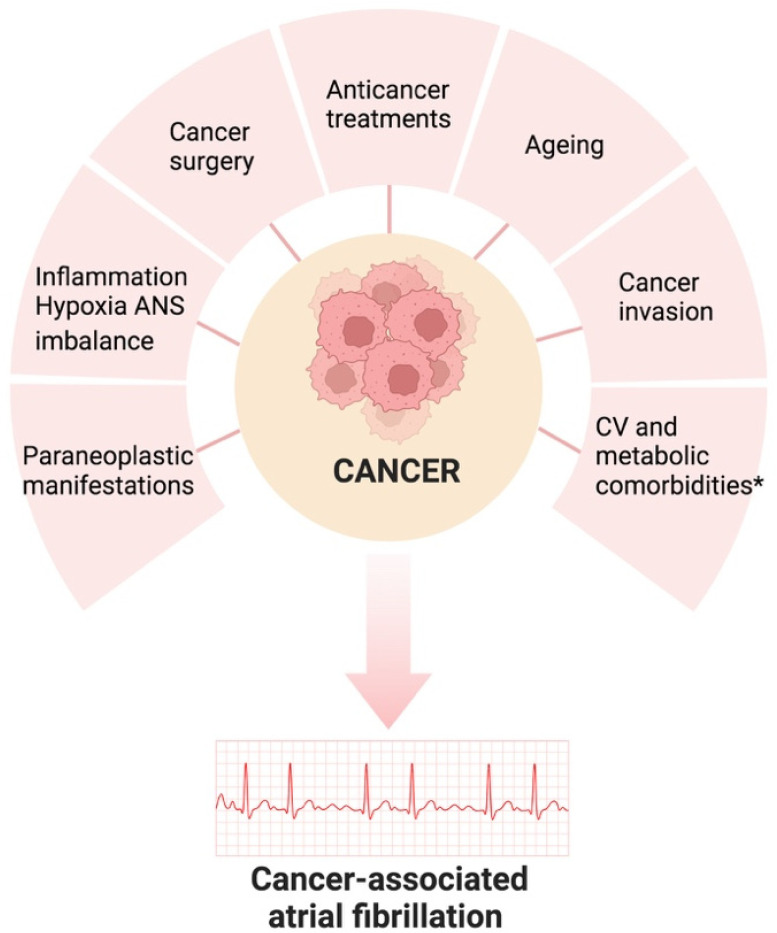
Pathogenesis of atrial fibrillation associated with cancer. ANS: autonomic nervous system; CV: cardiovascular; * obesity, hypertension, DM, CVDs (HF, VHD, IHD, cardiomyopathies, cardiac amyloidosis). Created with Biorender.com.

**Figure 2 cancers-15-05357-f002:**
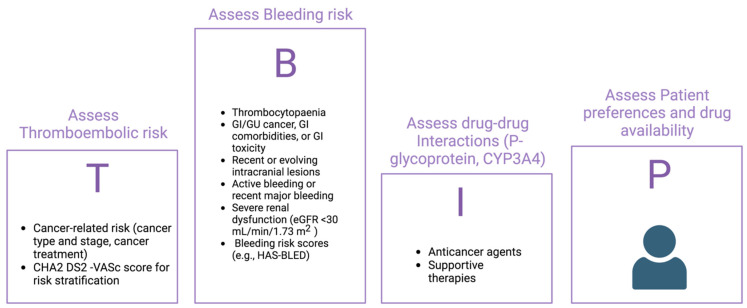
Structured approach to anticoagulation for atrial fibrillation in patients with cancer. AF: atrial fibrillation; CHA2DS2-VASc: congestive heart failure, hypertension, age ≥ 75 years (2 points), diabetes mellitus, stroke (2 points), vascular disease, age 65–74 years, sex category (female); eGFR: estimated glomerular filtration rate; GI: gastrointestinal; GU: genitourinary; HAS-BLED: hypertension, abnormal renal and liver function, stroke, bleeding, labile international normalized ratio, elderly, drugs or alcohol. Created with Biorender.com.

**Table 1 cancers-15-05357-t001:** Interactions between Bruton tyrosine kinase inhibitors (BTKis) and the drugs frequently used for the treatment of atrial fibrillation.

Drugs	Grade of Interaction	Consequences	Molecular Mechanisms
Diltiazem/Verapamil	Major	Increased plasma availability of ibrutinib (from 6- to 9-fold)	Diltiazem/verapamil are inhibitors of CYP450 3A4
Amiodarone/Dronedarone	Major	Increased plasma availability of ibrutinib (from 6- to 9-fold)	Amiodarone/verapamil are inhibitors of CYP450 3A4
Digoxin	Moderate	Increased plasma availability of digoxin	Ibrutinib inhibits P-glycoprotein
Factor Xa Inhibitors (Apixaban, Edoxaban, Rivaroxaban)	Moderate	Increased plasma availability of factor Xa inhibitors	Ibrutinib inhibits P-glycoprotein and induces CYP450 3A4
Direct Thrombin Inhibitor (Dabigatran)	Major	Increased plasma availability of dabigatran	Ibrutinib inhibits P-glycoprotein

**Table 2 cancers-15-05357-t002:** HAS-BLED score. A score of ≥3 indicates “high risk” and that some caution and regular review of the patient is needed. TTR: time in therapeutic range.

	Points	Condition
**H**—*Hypertension*	1	Systolic blood pressure > 160 mmHg.
**A**—*Abnormal Liver or Renal Function*	1 each	Abnormal renal function: dialysis, creatinine > 2.3 mg/dL, transplantation. Abnormal liver function: chronic hepatitis, cirrhosis, bilirubin > 2 ULN, ALT > 3 ULN.
**S**—*Stroke*	1	Previous history, particularly lacunar.
**B**—*Bleeding*	1	Recent bleeding, anemia, etc.
**L**—*Labile INR*	1	Unstable/high INR or TTR < 60%.
**E**—*Elderly*	1	Age > 65 years, extreme frailty.
**D**—*Drugs or Alcohol*	1 each	Prior alcohol or drug usage: history: ≥8 drinks/week; drugs: concomitant antiplatelets, NSAID use, etc.

**Table 3 cancers-15-05357-t003:** HEMORR2HAGES scores. Patients with scores of 0 or 1 were classified as low-risk, patients with scores of 2 or 3 were classified as intermediate-risk, and patients with scores of ≥ 4 were classified as high-risk.

	Points	
**H**—*Hepatic or Renal Disease*	1 each	
**E**—*Ethanol Abuse*	1	
**M**—*Malignancy History*	1	
**O**—*Older (Age > 75 Years)*	1	
**R**—*Reduced Platelet Count or Function*	1	Includes aspirin use and any thrombocytopenia or blood dyscrasia, like hemophilia.
**R**—*Rebleeding Risk*	2	
**H**—*Hypertension* (*Uncontrolled*)	1	
**A**—*Anemia*	1	Hgb < 13 g/dL for men; Hgb < 12 g/dL for women.
**G**—*Genetic Factors*	1	CYP 2C9 single-nucleotide polymorphisms.
**E**—*Excessive Fall Risk*	1	
**S**—*Stroke History*	1	

**Table 4 cancers-15-05357-t004:** Summary of available evidence on the use of NOACs for AF management in cancer patients. RCT: randomized controlled trial; VTE: venous thromboembolism; IS: ischemic stroke; SE: systemic embolism; GI: gastrointestinal; MI: myocardial infarction; CV: cardiovascular; NMCR: non-major clinically relevant.

Publication Year	Trial/Reference	Type of Evidence	Prospective/Retrospective	Number of Patients	Drug	Summary of Evidence
2019	ROCKET-AF [52]	Subgroup analysis of RCT	Prospective	640	Rivaroxaban	No efficacy or safety differences. Increased risk of bleeding.
2017	ARISTOTLE [54]	Subgroup analysis of RCT	Prospective	1236	Apixaban	Similar efficacy in preventing stroke and systemic embolism. No increase in major bleeding.
2018	ENGAGE AF-TIMI 48 [55]	Subgroup analysis of RCT	Prospective	1153	Edoxaban	Similar efficacy and safety.
2018	Shah S, et al. [53]	Administrative analysis	Retrospective	16,096	Various NOACs	Lower or similar rates of bleeding and stroke and a lower rate of incident VTE.
2022	Potter AS, et al. [60]	Single-center database analysis	Retrospective	1133	Various NOACs	Similar risks for cerebrovascular accident, gastrointestinal bleeding, and intracranial hemorrhage.
2020	Wu VC, et al. [68]	Administrative analysis	Retrospective	336	Various NOACs	Reduced IS/SE, major bleeding, and ICH compared to warfarin.
2019	Yasui T, et al. [69]	Single-center database analysis	Retrospective	127	Various NOACs	Similar rates of IS, SE, and major bleeding.
2018	Kim K, et al. [70]	Single-center database analysis	Retrospective	388	Various NOACs	NOACs associated with lower incidences of IS/SE, major bleeding, and all-cause mortality.
2017	Ording AG, et al. [71]	Administrative analysis	Retrospective	1809	Various NOACs	Similar risks of SE or bleeding in patients with and without cancer.
2021	Mariani MV, et al. [58]	Meta-analysis	Prospective/Retrospective	46,424	Various NOACs	NOACs associated with reduction in thromboembolic events and major bleeding.
2023	Tran E, et al. [72]	Single-center database analysis	Retrospective	58	Various NOACs	Evidence for management issues during chemotherapy.
2022	Parrini I, et al. [73]	Meta-analysis	Prospective/Retrospective	228,497	Various NOACs	NOACs showed better efficacy and safety outcomes than warfarin.
2021	Liu F, et al. [74]	Meta-analysis	Prospective/Retrospective	248,218	Various NOACs	Reduction in SE, VTE, and intracranial and GI bleeding. Same risk of IS, MI, CV death, all-cause death, major bleeding, and major or NMCR bleeding.

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
