# Peer review of "Current Data and Future Perspectives on Patients with Atrial Fibrillation and Cancer"

_cancers, 2023, doi:10.3390/cancers15225357_

Round 1

Reviewer 1 Report

Comments and Suggestions for Authors

1. Please add vomiting and diarrhea as causes of electrolyte shifts and atrial fibrillation. And also pain can be a cause of atrial fibrillation.

2. According to recent ESC guidelines on cardio-oncology, the use of NOACs in can- 283 cer patients with AF is broadly accepted in light of previous findings even if a clear pro- 284 spective evaluation is lacking. NOAC should be considered for stroke prevention in- 285 stead of LMWH and VKA in patients without significant drug-drug interactions, me- 286 chanical heart valves, or moderate-to-severe mitral stenosis

NOACs should not be considered in patients with gastrointestinal cancers due to increased risk of bleeding- add literature below

3. Please add information that in prognostic patients with 12-month survival, the left atrial appendage can be closed if NOAC/OAC is contraindicated.

References to add:

A practical approach to the ESC 2022 cardio-oncology guidelines. Comments by a team of experts: cardiologists and oncologists.

Leszek P, Klotzka A, BartuÅ› S, Burchardt P, Czarnecka AM, DÅ‚ugosz-Danecka M, Gierlotka M, KoseÅ‚a-Paterczyk H, Krawczyk-Ożóg A, Kubiatowski T, Kurzyna M, Maciejczyk A, Mitkowski P, Prejbisz A, Rutkowski P, Sierko E, SterliÅ„ski M, Szmit S, Szwiec M, Tajstra M, TyciÅ„ska A, Witkowski A, Wojakowski W, Cybulska-Stopa B.

Kardiol Pol. 2023 Sep 3. doi: 10.33963/v.kp.96840. Online ahead of print. PMID: 37660389

Author Response

R1: Please add vomiting and diarrhea as causes of electrolyte shifts and atrial fibrillation. And also pain can be a cause of atrial fibrillation.

A: We thank the reviewer for the suggestion. We have included vomiting, diarrhoea and pain among the possible causes of AF

R1:  According to recent ESC guidelines on cardio-oncology, the use of NOACs in cancer patients with AF is broadly accepted in light of previous findings even if a clear prospective evaluation is lacking. NOAC should be considered for stroke prevention instead of LMWH and VKA in patients without significant drug-drug interactions, mechanical heart valves, or moderate-to-severe mitral stenosis

NOACs should not be considered in patients with gastrointestinal cancers due to increased risk of bleeding- add literature below

A: We thank the reviewer for the suggestion, We expressed this very important concept in the section "Choice of anticoagulant therapy", page 9.

R1: We have included the possibility of left auricle closure in patients with contraindication to anticoagulant (paragraph: Choice of anticoagulant therapy)

A: We thank the reviewer for the suggestion. We have added the possibility of left auricle closure in patients with contraindication to anticoagulant in the paragraph “Choice of anticoagulant therapy”

R1: Please add information that in prognostic patients with 12-month survival, the left atrial appendage can be closed if NOAC/OAC is contraindicated.

A: We added a recent retrospective analysis regarding the outcome of cancer patients undergoing left auricular closure

References to add:

A practical approach to the ESC 2022 cardio-oncology guidelines. Comments by a team of experts: cardiologists and oncologists.

Leszek P, Klotzka A, BartuÅ› S, Burchardt P, Czarnecka AM, DÅ‚ugosz-Danecka M, Gierlotka M, KoseÅ‚a-Paterczyk H, Krawczyk-Ożóg A, Kubiatowski T, Kurzyna M, Maciejczyk A, Mitkowski P, Prejbisz A, Rutkowski P, Sierko E, SterliÅ„ski M, Szmit S, Szwiec M, Tajstra M, TyciÅ„ska A, Witkowski A, Wojakowski W, Cybulska-Stopa B.

Kardiol Pol. 2023 Sep 3. doi: 10.33963/v.kp.96840. Online ahead of print. PMID: 37660389

A: We added this reference in the text

Reviewer 2 Report

Comments and Suggestions for Authors

Dear authors,

Congratulations on your work. In my opinion your literature review is of high interest for the readers and on an extremely interesting and modern topic.

As this is more like a narrative literature review, I see no issues regarding its content structure. 

However, I still see some minor changes necessary in order to make it publishable. First of all try to add in the Introduction section data regarding co-adjuvant drugs and oncological treatments (e.g. doi:10.3990/medicina58091239). Also, the topic regarding patient preexisting disease, any other medical conditions (e.g.diabetes, ...) could influence AF development.

Regarding manuscript structure, please revie thoroughly the Instructions for authors section, as there are multiple flows in your manuscript from this point of view (e.g. rewrite References sectoon, keep template structure of the manuscript, also in the abstract,...)

In my opinion this manuscript should be revised and the final version should be re_revised.

Best regards,

Author Response

A: Dear reviewer, thank you for your opinion. As you suggested, we have introduced a small section on risk factors in the development of atrial fibrillation also in the general population. We also took advantage of your advice regarding the inclusion of co-adjuvant therapies, a cancer therapy and some evidence on interactions with various neoplasms in the introductory section.

We undertook a search and correction of inconsistencies regarding the bibliography, removing bibliography that was not necessary for the drafting of the paper.

Finally, we added a small summary paragraph.

Reviewer 3 Report

Comments and Suggestions for Authors

In the present manuscript, De Luca et al. analyzed the atrial fibrillation (AF) and cancer. The Authors therefore aimed at discussing novel etiological aspects underlying AF occurrence in cancer patients, to give advice on management aspects and shed light on future research scopes in this expanding field of cardio-oncology. They first evaluated the risk factors responsible for the high incidence of AF in cancer patients, including the overall pro-inflammatory state, chemotherapy and surgical resection. The Authors then focus on the therapeutic management of cancer patients with AF who undergo the same rhythm and frequency control strategies as the general population, but who may also suffer several side effects because of the interaction with chemotherapeutics and of their extreme fragility. The Authors also discuss the risks associated to anticoagulant therapy and the requirement to fully assess bleeding and stroke risk scores in this subpopulation. They also underline that, in large studies establishing the efficacy of direct oral anticoagulants (DOACs), cancer patients have been underrepresented.

Overall, the manuscript has been well written and provides a reliable source of information on the relationship between cancer and atrial fibrillation, which represents an increasing risk factor for cancer patients that oncologists must take in account to design a correct therapeutic approach. I only have some minor comments that should be addressed before the manuscript is fully suitable for publication.

Line 35: There is a typo here: (1;2).

Line 36: I would introduce here the expanded definition of AF

Line 74: There is a space missing here: (18).Many

Line 109: There is a type also here: recommended. (28)

Line 176: A type also here: in those without .

Line 210: There is a space missing here: patient.Vitamin

Line 234: For benefit of the readers, I would suggest to indicate also the percentage valure of 640 out of 14264

Line 298: A type here: these results. (69)

Author Response

 Dear reviewer, thank you for your suggestions. We have made the requested changes and corrected some errors in the text

Round 2

Reviewer 2 Report

Comments and Suggestions for Authors

Dear Authors,

The reviewed version of the manuscript is much more improved. I recommend it to be published in the current form. Congratulations for your efforts!

Author Response

Thank you.